# Preparation of CdTe/Alginate Textile Fibres with Controllable Fluorescence Emission through a Wet-Spinning Process and Application in the Trace Detection of Hg^2+^ Ions

**DOI:** 10.3390/nano9040570

**Published:** 2019-04-08

**Authors:** Zhihui Zhao, Cunzhen Geng, Xihui Zhao, Zhixin Xue, Fengyu Quan, Yanzhi Xia

**Affiliations:** State Key Laboratory of Bio-Fibers and Eco-Textiles, Shandong Collaborative Innovation Center of Marine Biobased Fibers and Ecological Textiles, Institute of Marine Biobased Materials, Qingdao University, Qingdao 266071, China; qdugcz@qdu.edu.cn (C.G.); zhaoxihui@qdu.edu.cn (X.Z.); xuezhixin@qdu.edu.cn (Z.X.); quanfengyu@qdu.edu.cn (F.Q.)

**Keywords:** fluorescence, textile fibres, alginate, CdTe, wet-spinning

## Abstract

Fluorescent textile fibres (FTFs) are widely used in many industrial fields. However, in addition to fibres with good fluorescence, fibres with excellent colour controllability, structural stability and appropriate mechanical strength still need to be developed. In this work, CdTe/alginate composite FTFs are prepared by taking advantage of the interactions between CdTe nanocrystals (NCs) and alginate macromolecules via a wet-spinning machine with a CaCl_2_ aqueous solution as the coagulation bath. CdTe NCs were chemically fixed in the fibre due to the interactions among surface ligands, macromolecules and coagulators (calcium ions), which ensured the excellent dispersity and good stability of the fibres. Förster resonance energy transfer (FRET) between NCs in the fibre was found to be restricted, which means that the emission colour of the fibres was totally controllable and could be predicted. Other properties of alginate fibres, such as flame retardance and mechanical strength, were also well preserved in the fluorescent fibres. Finally, FTFs showed good selectivity toward trace Hg^2+^ ions over other metallic ions, and the detection could be identified by the naked eye.

## 1. Introduction

Alginate is a kind of natural polysaccharide extracted from brown algae and brown seaweed. The structure of alginate is a linear chain composed of α-L-guluronic acid (G segment) and β-D-mannuronic acid (M segment) linked by 1,4-glycosidic bonds, with one carboxy group and two hydroxy groups on each glucose unit [1,2]. Sodium alginate (SA) is the most common form of alginate, and it is water-soluble and easy to process into different shapes [3,4]. When SA encounters divalent metal cations, the carboxy and hydroxy groups on the G segments will coordinate with the cations to form an insoluble “egg-box” cross-linking structure, which basically maintains its original shape [5]. This unique property enables the preparation of alginate textile fibres through a wet-spinning process, using SA as the spinning solution and divalent metal salt solutions as the coagulant [6,7]. Alginate textile fibres, which are important supplements for natural and modified textile fibres, such as cotton, wool, silk, and modified cellulose, are biocompatible, biodegradable, inexpensive, easy to prepare and can be widely used to make clothes, decoration materials and disposable non-woven medical fabrics [8]. Because land resources are currently being exhausted, the investigation and utilization of ocean-based alginate fibres are of great significance. However, the varieties and properties of alginate fibres are still not sufficient to meet the needs of different applications.

Among functional textile fibres, fluorescent textile fibres (FTFs) have attracted tremendous attention for their applications in decorations [9], sensors [10], illumination and display materials [11], and anti-counterfeiting materials for bills and stamps [12,13]. Various kinds of FTFs have been prepared. Ge et al. spun blends of fluorescent rare-earth materials and fibre-formed polymers to obtain fluorescent anti-counterfeiting fibres, but the emission colours of these materials were greatly affected by the environment and spinning processes [14]. Tansil et al. reviewed the preparation and application of luminescent silkworm silk fibres [10], and discussed the fabrication of fluorescent silk fibres by incorporating fluorescent CdTe and CdS nanocrystals (NCs) onto the surface of the fibres via electrostatic assembly [12] and γ-ray irradiation [15]. However, the incorporation of NCs onto the surface of a fibre, rather than internally, may result in lower composite stability, and worse, direct contact between human skin and NCs with heavy metals on the fibre surfaces increases the risk of pathopoiesia [16,17]. Thus, a variety of FTFs with bright colours, colour-controllable and colour-predictable emission, excellent stability and safety still need to be developed. In addition, FTFs using alginate fibres as a matrix retain the additional advantages of alginate and are thus worth investigating.

Considering the water-solubility of the SA used as the raw material for fibre growth, water-dispersed CdTe NCs were chosen as the fluorescent additives to prepare FTFs. CdTe NCs, as a kind of representative fluorescent semiconductor, have attracted significant attention over the last two decades. Water-soluble CdTe NCs have abundant advantages, such as high quantum efficiency, controllable emission colours, mono-dispersity in solution, adjustable surface properties and easy preparation [18]. To realize the practical applications, CdTe NCs usually must be incorporated into a polymer matrix [19,20]. However, NC/polymer composites in the form of macroscopic fibres, especially textile fibres, have rarely been reported.

In this report, we proposed a simple green method for generating high-quality CdTe/alginate composite FTFs by wet-spinning a blend of water-dispersed CdTe NCs and water-soluble alginate. First, the fluorescent and rheological properties of the blends of SA and different CdTe NCs were studied to choose the suitable CdTe NCs. Then, the wet-spinning process was conducted, and FTFs with different emission colours were prepared. The physical and chemical properties of the FTFs were studied, and details of FTF construction were investigated. Finally, a preliminary attempt was made to use the FTFs for trace mercury ion detection. Our research will provide a simple and economical way to generate composite textile fibres with excellent fluorescence properties.

## 2. Materials and Methods

### 2.1. Materials

Tellurium powder, thioglycolic acid (ThGA), mercaptoethanol (ME) and mercaptoethylamine (MEA) were all analytical grade from Sigma-Aldrich (Saint Louis, MO, USA). SA powder (Mw = 220 kDa and MWD = 1.5) was analysed with tri-detector analysis gel permeation chromatography (Malvern Viscotek TDA max); M/G = 1.05 was determined by Fourier transform infrared (FT-IR) spectroscopy analysis with a Nicolet 5700 spectrometer (Madison, WI, USA) FT-IR [21]. All other chemicals used in the experiment were from Sinopharm Chemical Reagent Co., Ltd. (Shanghai, China). Deionized water was used throughout the experiment.

### 2.2. Preparation of the CdTe-SA Spinning Dope (SD)

Water-soluble CdTe NCs with different surface ligands, including ThGA, ME and MEA, were prepared according to the traditional method [18]. Furthermore, the fluorescent colours of NCs were regulated by adjusting the reaction times according to the literature.

First, SA powers were dissolved in water at 50 °C to form a 5.0 wt% solution. Then, CdTe NC dispersions were blended with the SA solution (1:4 *v*/*v*) by vigorous stirring to produce the SD. The NCs listed above were all prepared to select the most suitable one for achieving a stable blending solution with excellent fluorescent and rheological properties. The prepared SDs were kept overnight to eliminate internal bubbles.

#### Measurements of SDs

Fluorescence measurements of SDs were recorded using a fluorescence spectrometer (Varian Cary Eclipse, Palo Alto, CA, USA).

Rheological analysis of the SDs was performed with a rotation rheometer (MCR30, Anton Paar, Graz, Austria). A plate fixture with a 50 mm diameter and a gap distance of 0.5 mm was used. The measurement temperature was 50 °C. For dynamic rheological analysis, stress sweeps were run with a constant strain of 5% to confirm the linear viscoelastic region.

### 2.3. Preparation of CdTe/Calcium Alginate (CA) FTFs via Wet Spinning

The formation of CdTe/CA FTFs was conducted with a homemade wet-spinning machine. Figure 1 shows a sketch of the wet-spinning process. SDs were extruded quantitatively by a metering pump and then passed through a spinneret into a coagulation bath. Fibres formed in the coagulation bath were then collected on a roller.

In our experiment, a 5.0 wt% CaCl_2_ aqueous solution was chosen as the coagulation bath. Here, CaCl_2_ was chosen due to its nontoxicity, low cost and excellent gelation ability with SA. During the coagulation process, Ca^2+^ ions exchanged with the counter Na^+^ ions of SA chains, and then coordinated with the hydroxyl and carboxyl groups to form egg-box structures [5], forming insoluble CA fibrous gels. Then, the as-spun fibres were washed thoroughly with water to remove excessive Ca^2+^ and vacuum freeze-dried. Pure alginate fibres were prepared using an SA solution as an SD, and FTFs were prepared using CdTe-SA-blended SDs.

#### Measurement of FTFs

The amount of CdTe NCs in FTFs was measured by a inductively-coupled plasma spectrometer (ICP, PerkinElmer Avio 200, Waltham, MA, USA). The morphologies of the fibres were obtained with a field emission scanning electron microscope (SEM, JEOL JSM-6700F, Tokyo, Japan), with a primary electron energy of 3 kV, and the samples were sputtered with a thin layer of Pt (with a thickness of 5 nm). Cross-sections were prepared by a Harrington slicer. For the slicing process, a bundle of fibres was clamped in the slicer, and the excess was removed. The cross-section was then fixed with collodion, the fixed section of the collodion was carefully scraped from the slicer, and the reverse side was observed by SEM. FT-IR was recorded as described above. The interior structures of the FTFs were recorded by observing a pre-prepared ultra-micro-cut sample with a transmission electron microscope (TEM, HITACHI H-7650, Tokyo, Japan).

Fluorescence spectra of the FTFs were recorded using a fluorescence spectrometer (Varian Cary Eclipse, Palo Alto, CA, USA). The absolute quantum yield (QY%) and fluorescence lifetime were measured on a fluorescence lifetime and steady-state spectrometer (Edinburgh Instrument, FLS 920, Edinburgh, Britain). Thermal stability was investigated by thermogravimetric analysis (TGA; PerkinElmer TGA7, Waltham, MA, USA) over a temperature range from 30–700 °C at a rate of 10 °C/min. The flame retardant properties of the fibres were recorded by measuring the limiting oxygen index (LOI) using an LOI determinator (Juefeng HC-2, Nanjing, China). The LOI sample was prepared by packaging the fibres with a piece of tinfoil and then pressing them into a film with the dimensions 100 × 37 × 2 mm. The mechanical strengths of the fibres were confirmed by a Favimat-Airobot Single-Fibre Testing Machine (Textechno Company, Mönchengladbach, Germany).

### 2.4. Detection of Hg^2+^ Ions

Solutions of BaCl_2_, CaCl_2_, AlCl_3_ and FeCl_3_ were prepared at a concentration of 5 μM, and a solution of HgCl_2_ was prepared at a concentration of to 1μM. Then, 0.01 g FTF samples were dipped into 10 mL of the solutions for 5 min and then removed. Hg^2+^ could be visibly detected by the naked eye with the aid of a simple UV lamp (365 nm).

## 3. Results and Discussion

### 3.1. Selection of CdTe NCs with the Most Appropriate Surface Ligands to Prepare SD

In our experiment, water-soluble CdTe NCs with ThGA, ME and MEA as surface ligands were separately blended with SA solutions (1:4 *v*/*v*). The outer functional groups of the ThGA-, ME- and MEA-modified NCs were carboxyl, hydroxyl and amino groups, respectively, resulting in different interactions with SA during blending. The fluorescence properties of all blends were tested to determine the most stable blend. The measurement results and discussion are shown in the first part of the Appendix A. Based on the experimental results shown in the ESM, the ThGA-CdTe/SA blend was chosen as the SD. Then, the rheological properties of ThGA-CdTe/SA were tested, as shown in the second part of the ESM. The results showed that the addition of CdTe NCs slightly decreased the viscosity of the SA solution, but simultaneously increased the viscous proportion, which ultimately increased the spinning ability of SA. Based on the above research, a ThGA-CdTe/SA blend was prepared as the SD, in which the concentration of CdTe NCs was approximately 0.5 wt%.

### 3.2. Preparation, Morphology and Structure of FTFs

During the spinning process, fibres formed in the CaCl_2_ coagulation bath, and Ca^2+^ ions exchanged with the Na^+^ ions on the polymer chains and then coordinated with the hydroxyl and carboxyl groups to form egg-box structures [5,22]. The thin stream of polymer gelled into primary fibres, and the nature of the polymer changed from water-soluble SA to water-insoluble CA. CA textile fibres (CTFs) were collected on the roller, as shown in Figure 1. FTFs were prepared in the same way by using ThGA-CdTe/SA SDs instead of the pure SA SD. The amount of CdTe in FTFs was tested to be 2415 ppm by ICP, which was near half the amount of CdTe in the spinning dope (0.5 wt%). The loss of CdTe occurred in the coagulation process. NCs at the interface of the spinning dope and the coagulation bath diffused outside into the bath before Ca^2+^ diffused inside and coagulated the fibres.

The morphologies of CTFs and FTFs were characterized using SEM. Figure 2a,b show the plane view images of CTFs and FTFs, respectively. Both samples had rough surfaces with grooves and convex morphologies. The width of the FTFs was approximately 50 μm, while that of the CTFs was only 15 μm. This difference might be because the lower viscosity of ThGA-CdTe/SA SDs resulted in an increase in the number of SDs passing through the spinneret during spinning. The FTFs on the substrate in Figure 2b appeared flat rather than round, which was confirmed by the cross-section SEM image in Figure 2c. The fibres at the edges showed obviously flat shapes, suggesting that the thin stream of SD diffused anisotropically in the coagulation bath, due to the formation of corn-husk-like structures during the spinning process, which was induced by the faster moulding on the outer surface of the fibre than in the interior in the coagulation bath. The inner composition of FTFs was studied by TEM. Figure 2d shows that CdTe NCs in FTFs were well dispersed in the fibres and separated from each other, and no aggregation was observed. However, the final NCs (approximately 10–13 nm) were larger than the original NCs (4–5 nm) because of the existence of surface polymers instead of the original small ligands.

FT-IR was used to further investigate the structure of CTFs and FTFs (Figure 2e). Differences were observed mainly between the fibres and the SA powders, while the spectra of FTFs and CTFs showed little difference. Compared with the peaks in the spectra of the SA powders, the peaks attributed to O-H stretching vibrations at 3400 cm^−1^ and bending vibrations at 1031 cm^−1^ in the spectra of CTFs and FTFs became less intense and broader, the peak attributed to COOH stretching vibrations at 1609 cm^−1^ shifted to 1627 cm^−1^ and the peak at 1423 cm^−1^ decreased. These changes showed that both the COOH and OH groups of the alginate chains coordinated with Ca^2+^ ions to form “egg-box” structures. The peak at 2930 cm^−1^ was attributed to the stretching vibration of CH groups on the polysaccharide units of alginate, which was also confined by the formation of an “egg-box” structure, resulting in a decrease in the peak. Since the COOH groups of ThGA-CdTe NCs also participated in the coordination with Ca^2+^ ions in FTFs, there was no significant difference between the spectra of CTFs and FTFs.

### 3.3. Photoluminescence (PL) Properties and Possible Composition of FTFs

Figure 3a shows optical images, under ambient light and UV light, of FTFs of three different colours that were formed by the wet spinning of ThGA-SA SDs blended with ThGA-CdTe NCs of different colours. As shown in the figure, the pure CTFs were white. After the wet-spinning process, the original white fibres became homogeneous red, pink, and buff fibres, according to the different CdTe NCs in the SDs, thus providing a new method of generating coloured alginate textile fibres. As shown in the lower images of Figure 3a, after illumination by UV light, the FTFs showed strong orange, yellow and green fluorescence emissions, while no emission could be observed from the CTFs. Fluorescence was observed throughout the composite fibres, and the colours were very bright and uniform, suggesting excellent dispersion of CdTe NCs in the fibre. PL spectra show the emission of fluorescence fibres with different colours (Figure 3b). To our surprise, despite the different colours, the emission peaks of all FTFs were at almost the same position as the corresponding SDs. Therefore, the emission colours of the FTFs prepared in this study are completely predictable, which would greatly increase the controllability of the FTFs in practical production and applications. It is known that the emission colours of CdTe NCs often redshift to longer wavelengths when they are incorporated a solid matrix, due to the Förster resonance energy transfer (FRET) effect, especially when NCs were used without ligand exchange, surface passivation or other modifications [23]. To understand the change of interactions between NCs and the matrix (first SD and then CA fibres), FTFs doped with two kinds of CdTe NCs (green and yellow CdTe NCs) were prepared, and the non-shifting emission phenomenon was detected. As shown in Figure 3c, the PL spectra of the FTFs and the SD were quite different: a curve with two emission peaks, at 545 nm and 576 nm, was observed in the PL spectrum of the FTFs, which corresponded precisely to the peaks of the original NCs, while only one, much broader, emission peak appeared at 555 nm in the SD spectrum, due to the FRET effect between freely moving NCs in the SD. However, the preservation of the PL emission of the FTFs indicated that the FRET effect was prevented, which is in accordance with the non-shifted emissions of the unicolour FTFs discussed above. The evidence indicated that NCs in the FTFs were fixed and isolated from each other, which is in accordance with the TEM image shown in Figure 2d. The QY% of the FTFs was measured to be approximately 8.95%. The fluorescence decay profiles of the FTFs were generally fitted with multi-exponential components, and the average lifetime was calculated to be 27.2 ns. τ_1_ in Figure 3d represents the original intrinsic fluorescence, while τ_2_ results from the surface emission. The emergence and relatively large value of τ_2_ indicated that the surface properties of the CdTe NCs in the FTFs changed.

Based on the above results, the proposed composition of the FTFs is shown in Figure 3e: polymer chains were cross-linked into networks by Ca^2+^ cations during the coagulation process, and simultaneously, the NCs were confined or located in the polymer networks, maintaining their excellent dispersity in the fibres, as in the SD thin stream. Moreover, the outer functional groups on the CdTe NCs were carboxyl groups, which could also participate in coordination with Ca^2+^, and further fixed the NCs in their positions. In other words, the CdTe NCs were thought to be chemically imbedded in the fibres. This chemical bonding structure was further tested by soaking the FTFs in water, and indeed, the CdTe NCs were unable to escape from the fibres. Since CA severely swelled in water, up to several times its original size, if the CdTe NCs were simply physically embedded in the fibre, NCs would be prone to diffuse from the fibre into the water. However, no fluorescence was observed in the water after soaking for 2 days, as confirmed by the PL test. As shown in Figure 3f, the fluorescent fibres still emitted brightly in water, with clear shapes and boundaries. Moreover, the good chemical stability of the NCs in the fibre matrix effectively avoided possible contamination by CdTe NC leaks, greatly enhancing the safety of the FTFs in daily usage.

### 3.4. Thermal Stability, LOI and Mechanical Strength of FTFs

TGA was conducted to determine the thermal stability of the fibres (Figure 4). The CTF and FTF curves almost overlapped because of the small amount of CdTe NCs in the FTFs, which also indicated that the addition of CdTe NCs did not change the main composition of the fibre. The weight loss before 200 °C mostly arose from the loss of water. When heated to above 200 °C, the fibres started to decompose. When heated to 700 °C, the fibres completely decomposed. The thermal degradation residues of pure CA fibres were approximately 40.0% of the original fibres and were mainly composed of calcium oxide and calcium hydroxide [24]. From the enlarged image of Figure 4b, it is clear that the amount of FTF residues was approximately 1 wt% larger than the amount of CTF residues, which is larger than the additive amount of CdTe NCs in the composite fibres. This result might be because of the chemical interaction between NCs and the polymer. Therefore, the thermal stability of the FTFs was better than that of the CTFs.

An outstanding property of the CTFs is their intrinsic flame retardant property: the residues that formed after exposure to a flame could act as a barrier or insulating surface to prevent further decomposition of the fibres [25]. The coordination between the polymer chains with Ca^2+^ ions played a key role in the flame retardance. To investigate whether the addition of CdTe NCs influenced the flame retardance of the fibres, the LOI values of both fibres were recorded. The LOI for FTFs was 35.0, while the value for CTFs was 39.0. In FTFs, in addition to polymer chains, CdTe NCs also participated in the coordination with Ca^2+^, partly influencing the coordination structures formed between Ca^2+^ and polymer chains, leading to a decrease in the LOI value.

The mechanical properties of fibres are one of the most important properties in most applications, since fibres are usually used under various forces. Here, we investigated the tensile properties of the two fibres. Since the strengths of single fibres varied widely, the tensile tests required multiple single fibres in order to obtain an average value. In the textile industry, a greater number of fibres used in the test achieves better results, and usually no fewer than 30 single fibres are used. Here, for each sample, approximately 60 single fibres were tested to obtain an average value. All tensile curves are shown in Figure 5. The fineness of CTFs was tested to be 5.97 dtex and FTFs was 9.18 dtex. The average breaking strength and the average elongation at break were calculated to be 0.84 cN/dtex, and approximately 3.92% for CTFs and 1.19 cN/dtex and 13.95% for FTFs. Apparently, the addition of CdTe NCs to the fibres greatly improved their tensile properties. Nanoparticles embedded in polymers are often found to affect the mechanical properties of polymers. In general, the aggregation or weak adhesion of nanoparticles to a matrix will lead to the degradation of mechanical properties, while good dispersity and strong adhesion will improve mechanical properties [26]. As we discussed before, in the FTFs, CdTe NCs were uniformly dispersed and chemically embedded in the CA matrix. Furthermore, CdTe NCs might serve as sites of cross-linking in CA networks, thus, the tensile properties were greatly enhanced by the addition of CdTe NCs. Moreover, the strength and elongation at break values of FTFs were consistent with those of commercial viscose fibres, which were approximately 1.15 cN/dtex and 14.39% [27], respectively, indicating FTFs have the potential to be used in the textile industry.

### 3.5. Detection of Hg^2+^ Using FTFs

Hg^2+^ ions are highly toxic to humans and are one of the most threatening pollutants in our environment [28]. The detection of Hg^2+^ ions is often limited by interference from other metal ions and the usage of large and expensive instruments. In addition to potential applications as anti-counterfeiting materials in textile and other fields, FTFs can also be used as rough but quick visual indicators for the detection of trace amounts of Hg^2+^ ions. The process can be achieved with the aid of a simple UV lamp, without a professional fluorometer. The fluorescence of CdTe NCs can be quenched by the interaction between NCs and Hg^2+^ ions via an excited state electron transfer mechanism [29,30]. The FTFs also showed excellent and selective fluorescence toward Hg^2+^ compared to other metal ions. As shown in Figure 6, upon exposure to UV light, the green FTFs were quenched only when dipped in the Hg^2+^ solution. Although the original concentrations of other metal ions were five times larger than that of Hg^2+^ ions, the FTFs maintained their green luminescence. The PL quenching of fibres with Hg^2+^ was observed within several seconds of dipping, and could be distinguished by the naked eye. Another advantage of this method was that compared to other CdTe NC-based probes, the fibres were much easier to recycle and handle, avoiding further contamination as much as possible.

## 4. Conclusions

In summary, we fabricated FTFs by incorporating water-dispersed CdTe NCs into alginate fibres. With the aid of surface carboxyl groups, TGA-capped CdTe NCs were well dispersed in the CdTe/SA SD and further combined with CA macromolecules to form “egg-box” structures during the wet-spinning process. Benefitting from the fast solidification of the fibres and the interactions between NCs and CA, the fluorescent colours of CdTe NCs were well preserved in FTFs and could be easily changed by using NCs of different colours. In addition, the confinement of NCs in the polymeric networks was thought to result from chemical interactions, which protected the NCs from slipping out of the fibres and guaranteed their safety in daily use. The composite fibres also maintained the thermal stability and flame retardance of the CA fibres. The mechanical strengths of the fibres were greatly enhanced after incorporating CdTe NCs, with values of 186.67 MPa for the breaking strength and 13.95% for the elongation at break, indicating that the fibres were suitable for use as textile fibres. In addition, the FTFs were used as quick visual indicators for the detection of trace amounts of Hg^2+^.

## Figures and Tables

**Figure 1 nanomaterials-09-00570-f001:**
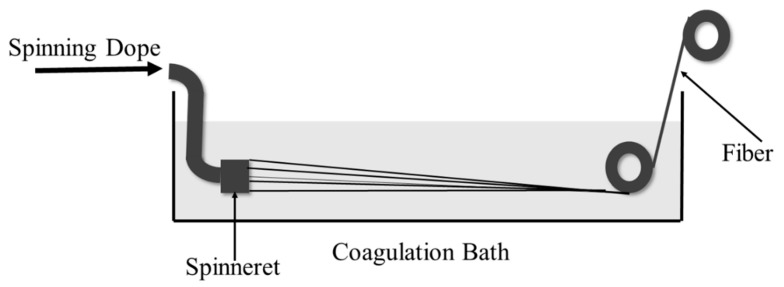
Schematic of the wet-spinning process for alginate fibre formation.

**Figure 2 nanomaterials-09-00570-f002:**
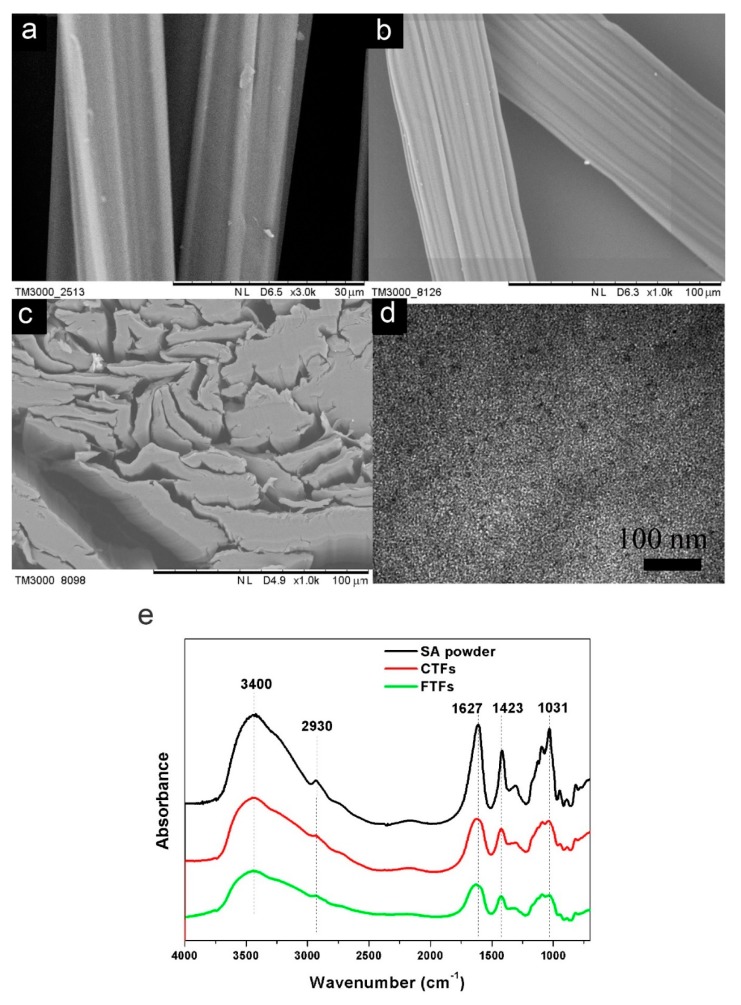
Plane view SEM images of (**a**) Calcium alginate textile fibres (CTFs) and (**b**) fluorescent textile fibres (FTFs), (**c**) cross-section SEM image of FTFs, (**d**) TEM image of FTFs and (**e**) FT-IR spectra of sodium alginate (SA) power, CTFs and FTFs.

**Figure 3 nanomaterials-09-00570-f003:**
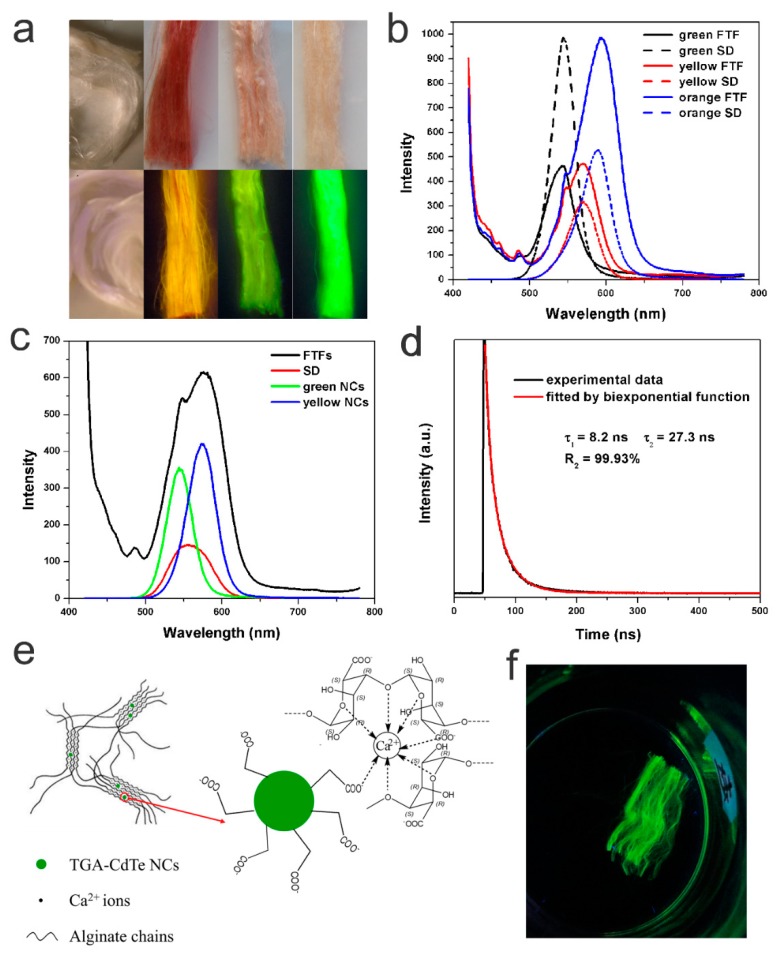
(**a**) Optical images of CTFs and FTFs with different colours under ambient light (upper) and excited by 365 nm UV light (lower). (**b**) Photoluminescence (PL) spectra of FTFs and corresponding SDs. (**c**) PL spectra of FTFs doped with two kinds of CdTe NCs. (**d**) Fluorescence decay profiles of FTFs with an excitation wavelength 365 nm. (**e**) Schematic illustration of the composition of FTFs. (**f**) Optical image of FTFs soaking in water for 2 days.

**Figure 4 nanomaterials-09-00570-f004:**
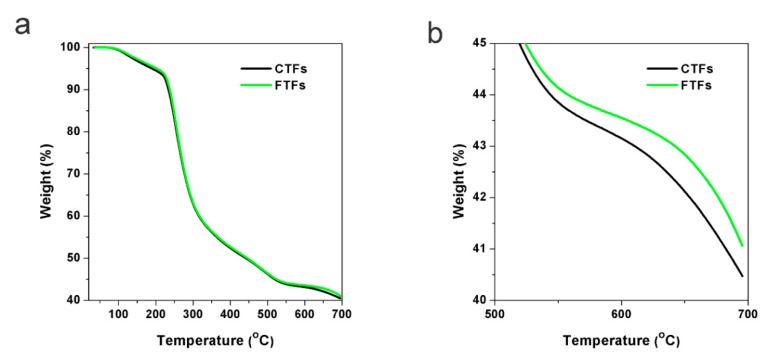
Thermogravimetric analysis (TGA) spectra of CTFs and FTFs: (**a**) the whole weight loss process; (**b**) magnified spectrum of the partial weight loss process.

**Figure 5 nanomaterials-09-00570-f005:**
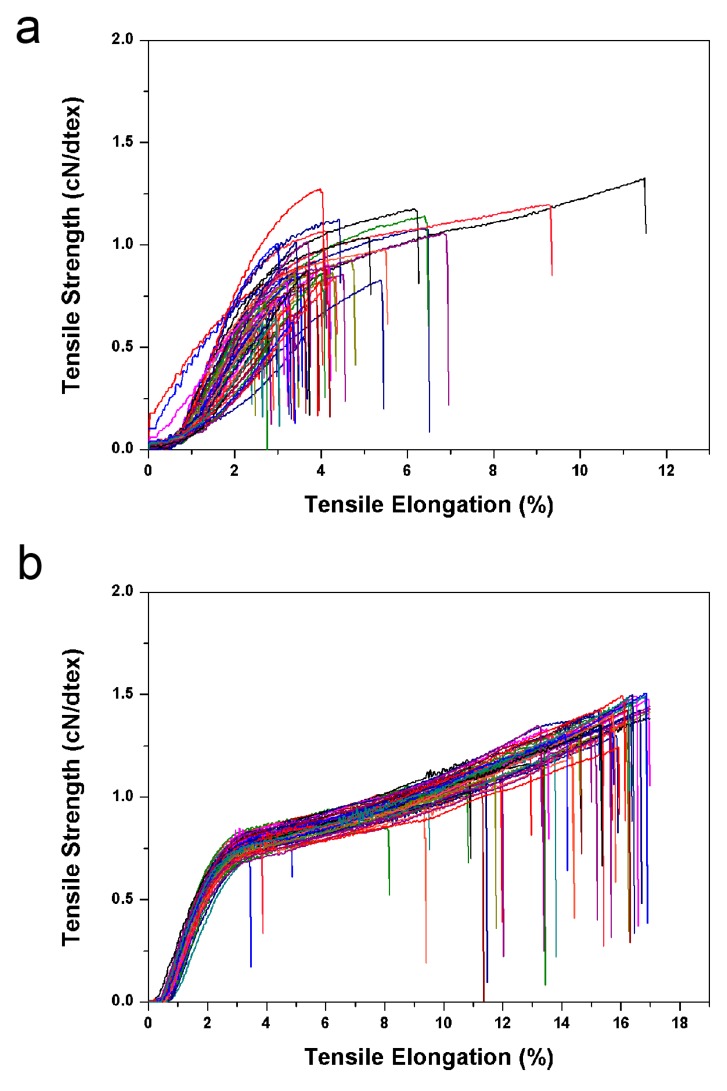
Tensile properties of (**a**) CTFs and (**b**) FTFs. Both fibres were tested 60 times to obtain average values.

**Figure 6 nanomaterials-09-00570-f006:**
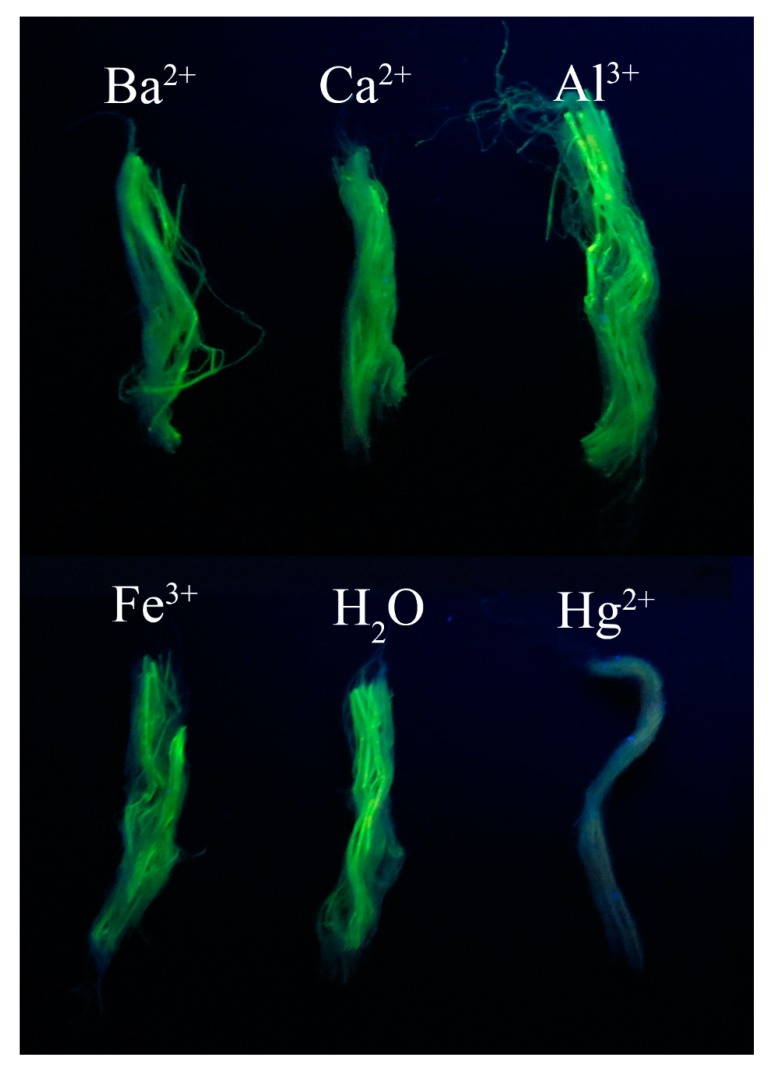
Optical images of green FTFs excited by a UV lamp (365 nm) after doping in different metal ion solutions. The concentrations were 5.0 μM for Ba^2+^, Ca^2+^, Al^3+^, and Fe^3+^ and 1.0 μM for Hg^2+^.

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
