# Peer review of "Preparation of CdTe/Alginate Textile Fibres with Controllable Fluorescence Emission through a Wet-Spinning Process and Application in the Trace Detection of Hg^2+^ Ions"

_nanomaterials, 2019, doi:10.3390/nano9040570_

Reviewer 1 Report

This paper is about the preparation and characterization of fluorescent CdTe/alginate textile fibres that can be deployed for the detection of Hg2+ ions in solution. The standard of the practical work is good but there are a number of observations that the authors should have made in order to present a more convincing case for the claims they make. In particular the use of TEM could have been further exploited to clearly show that CdTe is embedded in the fibre by using energy dispersive spectroscopy available on most TEM instruments. High angle annular bright field mode would have shown a better localization of the CdTe in the fibres. The absence of fluorescence in the solution when the fibres are soaked in water is no confirmation that CdTe are still in the fibres. What is the rationale for the selection of Al3+, Ba2+, Ca2+, and Fe3+ as potential interferences in the determination of Hg2+.  In the absence of confirmation by FTIR measurements of the reaction between CdTe and the COOH could other techniques have been used to produce evidence that the reaction had occurred? The absence of the evidence of a chemical reaction invalidates the point made in the conclusion.

Minor Comments

Line 24, should read: …..by the naked eye.

Line 56, delete obtain and replace by retain.

Line 57, delete researching and replace with investigating.

Line 69, delete proper and replace with suitable.

Line 158, Figure 2, label the bands with the appropriate wavenumbers.

Line 159 the part of the legend referring to Figure 2e is missing.

Author Response

Dear Professor:

Thank you for your comments concerning our manuscript. Those comments are all valuable and very helpful for revising and improving our paper, as well as the important guiding significance to our researches. We have studied comments carefully and have made correction which we hope meet with approval. Revised portion are marked in red in the paper. The main responses to the comments are as flowing:

Comment 1: The use of TEM could have been further exploited to clearly show that CdTe is embedded in the fibre by using energy dispersive spectroscopy available on most TEM instruments. High angle annular bright field mode would have shown a better localization of the CdTe in the fibres.
Response:

  We appreciate the Reviewer’s precious suggestion. EDS is a good method for qualitative and quantitative analysis of substances. However, we have tried to test our sample with EDS, but didn’t obtain a clear result. It may because the size of CdTe NCs were only 4-5 nm and separated from each other in FTFs, which made them very difficult to be detected by EDS. At present, the localization of CdTe NCs was determined from the changes in contrast in the TEM image (Fig. 2d). What is more, we measured the amount of CdTe NCs in FTFs quantitatively and the results were added in Section 3.2.

Comment 2: The absence of fluorescence in the solution when the fibres are soaked in water is no confirmation that CdTe are still in the fibres.
Response:

We deduced that CdTe NCs were still located in the fibres based on the following reasons:

First, compared the fibres in Fig. 3f with Fig. 3a, FTFs remained the original fluorescent color after soaking in water for 2 days. Second, as we discussed in Line 236 and 237 in the manuscript,  PL results showed that none fluorescence was detected in the water. Since the fluorescence of CdTe NCs were unlikely to be quenched during the process, we believed that CdTe NCs were still in the fibres.

Comment 3: What is the rationale for the selection of Al3+, Ba2+, Ca2+, and Fe3+ as potential interferences in the determination of Hg2+.

Response:

The selection detection of Hg2+ from other metal ions relied on the special excited state electron transfer of CdTe NCs with Hg2+, which will lead to the quenching of fluorescence. This special property of CdTe NCs have been used to specifically detect Hg2+ as reported in other researches (ref. 28-30 in our original article). Line 291-293 in the manuscript simply stated the principle.

Comment 4: In the absence of confirmation by FTIR measurements of the reaction between CdTe and the COOH could other techniques have been used to produce evidence that the reaction had occurred?
Response:

We are sorry for the inaccurate discussion of the comparison of FTIR spectra between CTFs and FTFs in the original article, which may have led to the misunderstanding of the Reviewer.

In fact, the chemical reaction we believed occurred during the spinning process was between CdTe NCs and Ca2+ in the coagulation bath. As we have discussed in section 3.1 and the ESI, CdTe NCs bearing ThGA (abbreviated as TGA-CdTe) rather than ME-CdTe and MA-CdTe was chosen to fabricate FTFs, which has the same carboxyl groups on the surface with the macromolecular chain of sodium alginate. During spinning, COO- groups on both surfaces of ThGA-CdTe NCs and the alginate chains reacted with Ca2+ in the coagulation bath at the same time, and finally resulted in the crosslinking networks of alginate and ThGA-CdTe NCs as shown in Figure 3e, which was the microstructure of FTFs.

Actually, the properties of surface groups of CdTe NCs did effect the experiment. As we discussed in the ESI, if CdTe NCs bearing amine groups on the surface (MA-CdTe) was blended with sodium alginate solution directly, hydrogel of MA-CdTe/alginate would generated. In addition, if Ca2+ was added directly into ThGA-CdTe dispersion, CdTe NCs would precipitate, which also confirmed the reaction between ThGA-CdTe and Ca2+.

Since both ThGA-CdTe NCs and alginate reacted with Ca2+ during spinning, the spectra of CTFs and FTFs would remain the same. The inaccurate description of this part of discussion (Line 183-186) in the original manuscript have been revised as follows: “Since the COOH groups of ThGA-CdTe NCs also participated in the coordination with Ca2+ ions in FTFs, there was no significant difference between the spectra of CTFs and FTFs.”

Comment 5: Minor Comments

Line 24, should read: …..by the naked eye.

Line 56, delete obtain and replace by retain.

Line 57, delete researching and replace with investigating.

Line 69, delete proper and replace with suitable.

Line 158, Figure 2, label the bands with the appropriate wavenumbers.

Line 159 the part of the legend referring to Figure 2e is missing.

Response:

We are sorry for our incorrect writing, and corrections have been made according to the Reviewer’s comments in the revised manuscript.

Special thanks to you for your good comments.

Reviewer 2 Report

Review of nanomaterials-465640

This manuscript describes the in situ incorporation of surface modified CdTe particles into alginate fibers during the fiber coagulation process from aqueous solutions. The authors characterized the obtained fibers with respect to morphology, mechanical and thermal and optical properties and partially composition.

The fibers are fluorescent due to the incorporated semi-conductor particles and Hg2+ ions quench this fluorescence. It is proposed that the fibers can be used as sensor to detect mercury.

I think the manuscript is well written and clearly presented. The results on the fiber properties are in my opinion very interesting for the scientific community.

I suggest to accept this manuscript for publication after a minor revision. My detailed comments:

1.       General: it would be good to have data on the chemical composition and the amount of incorporated CdTe from elemental analysis. How does the amount of incorporated metals depend on the spinning parameters and dope composition?

2.       Line 232: similar to the previous comment, is there any quantifiable data available to show that the leaching of particles is very low or non-existent?

3.       Line 266: to my knowledge it is very common in textile industry to also test single fibers

4.       Line 278: what is the fineness of the fibers? Textile industry expresses this as e.g. cN/tex to compare the strength of fibers with different fineness. This data should be added.

5.       Figure 6: what is the limit of detection and is there a linear correlation between Hg2+ concentration and quenching?

6.       Supplementary info. Figure S3, caption should match the curve legend.

Author Response

Dear Professor:

Thank you for your comments concerning our manuscript. Those comments are all valuable and very helpful for revising and improving our paper, as well as the important guiding significance to our researches. We have studied comments carefully and have made correction which we hope meet with approval. Revised portion are marked in red in the paper. The main responses to the comments are as flowing:

Comment 1: it would be good to have data on the chemical composition and the amount of incorporated CdTe from elemental analysis. How does the amount of incorporated metals depend on the spinning parameters and dope composition?

Response:

  We appreciate the Reviewer’s precious suggestion. The amount of CdTe in FTFs was tested to be 2415 ppm by ICP, which was near half the amount of CdTe in the spinning dope (0.5 wt%). We thought the loss of CdTe occurred in the coagulation process. NCs at the interface of the spinning dope and the coagulation bath diffused outside into the bath before Ca2+ diffused inside of the SD and coagulate the fibre. This part of discussion have been added in Section 3.2.

  The loss of CdTe was hard to be eliminated by adjust the spinning parameters. Reducing the volume of the coagulation bath and the drafting rate may reduce the loss of particles. Increasing the proportion of NCs in the spinning dope will increase the amount of NCs in the fibres, but at the same time also increase the loss ratio. The amount of CdTe in FTFs was tested to be 4277 ppm when the amount of CdTe in the SD was 1 wt%.

Comment 2: Line 232: similar to the previous comment, is there any quantifiable data available to show that the leaching of particles is very low or non-existent?

Response:

  We believed that none CdTe NCs were leached from the fibres based on the following reasons: First, compared the fibres in Fig. 3f with Fig. 3a, FTFs remained the original fluorescent color after soaking in water for 2 days. Second, PL measurement is a highly sensitive tool to investigate the change of microenviroment of the NCs. as we discussed in Line 236 and 237 in the manuscript, PL results showed that none fluorescence was detected in the water. Since the fluorescence of CdTe NCs were unlikely to be quenched during the process, we believed that there was none CdTe NCs in water.

Comment 3: Line 266: to my knowledge it is very common in textile industry to also test single fibers

Response:

  There is a little confusion in the paper. In our experiment, the mechanical strengths of the fibres did confirmed by a Single-Fibre Testing Machine. In fact, we tested the mechanical strengths of 60 single fibre to obtain the average value. The inaccurate expression in the original manuscript has been changed.

Comment 4: Line 278: what is the fineness of the fibers? Textile industry expresses this as e.g. cN/tex to compare the strength of fibers with different fineness. This data should be added.

Response:

  The data of fineness for both fibres have been added in Section 3.4.

For tensile strengths, our original results was obtain in “cN/dtex”, and it has been converted into the general unit of materials “MPa” according to the equation: 1 cN/dtex=98*r. According to the Reviewer’s suggestion, the unit of the tensile strengths have been changed to cN/dtex in the revised manuscript.

Comment 5: Figure 6: what is the limit of detection and is there a linear correlation between Hg2+ concentration and quenching?

Response:

  It have been well studied that the fluorescence of CdTe NCs can selectively quenched by Hg2+ ions (Ref. 28-30). A general rule is that the degree of fluorescence quenching has a linear relationship with the concentration of Hg2+. By using BAS-capped CdTe dispersion, the limit of detection was reported as low as 1´10-9 M. However, these studies required the usage of professional fluorescence spectrometers.

  In our study, the detection of Hg2+ was achieved by the naked eyes instantaneously. Due to the limitation of human eyes, the detection limit can only be achieved to 1.0 mM by now. Below this concentration, the quenching will still happen, but become slow and indistinguishable. What is more, our detection is usually completed in a short time, because the exchange of Ca2+ in fibres with ions to be detected may happen gradually, influencing the fluorescence of CdTe NCs.

Comment 6: Supplementary info. Figure S3, caption should match the curve legend.

Response:

Corrections have been made according to the Reviewer’s comment in the revised manuscript.

Special thanks to you for your good comments.

Round  2

Reviewer 1 Report

The authors have addressed the points raised by the reviewer.

Reviewer 2 Report

The authors made appropriate changes to the manuscript and it can be accepted in its present form.